# COVID-19 Vaccine Acceptance among Social Media Users: A Content Analysis, Multi-Continent Study

**DOI:** 10.3390/ijerph19095737

**Published:** 2022-05-08

**Authors:** Ramy Shaaban, Ramy Mohamed Ghazy, Fawzia Elsherif, Nancy Ali, Youssef Yakoub, Maged Osama Aly, Rony ElMakhzangy, Marwa Shawky Abdou, Bonny McKinna, Amira Mohamed Elzorkany, Fatimah Abdullah, Amr Alnagar, Nashwa ElTaweel, Majed Alharthi, Ali Mohsin, Ana Ordóñez-Cruickshank, Bianca Toniolo, Tâmela Grafolin, Thit Thit Aye, Yong Zhin Goh, Ehsan Akram Deghidy, Siti Bahri, Jarntrah Sappayabanphot, Yasir Ahmed Mohammed Elhadi, Salma Mohammed, Ahmed Nour El-Deen, Ismail Ismail, Samar Abd ElHafeez, Iffat Elbarazi, Basema Saddik, Ziad El-Khatib, Hiba Mohsin, Ahmed Kamal

**Affiliations:** 1Department of Instructional Technology and Learning Sciences, Utah State University, Logan, UT 84321, USA; ramy.shaaban@usu.edu; 2Tropical Health Department, High Institute of Public Health, Alexandria University, Alexandria 21561, Egypt; ramy_ghazy@alexu.edu.eg; 3Epidemiology Department, High Institute of Public Health, Alexandria University, Alexandria 21561, Egypt; marwa.shawky@alexu.edu.eg (M.S.A.); samarabdelhafeez.epid@gmail.com (S.A.E.); 4Department of Communications Media, Indiana University of Pennsylvania, Indiana, PA 15705, USA; n.s.ali@iup.edu (N.A.); youssefyakoubb@gmail.com (Y.Y.); 5Nutrition Department, High Institute of Public Health, Alexandria University, Alexandria 21561, Egypt; hiph.magedaly@alexu.edu.eg; 6Faculty of Medicine, Alexandria University, Alexandria 21545, Egypt; ronyibrahim13@hotmail.com; 7Institute of Tropical Medicine and International Health, Charité—Universitätsmedizin Berlin, 13353 Berlin, Germany; bonny-dowd.mckinna@charite.de (B.M.); ana-magdalena.ordonez-cruickshank@charite.de (A.O.-C.); yong-zhin.goh@charite.de (Y.Z.G.); 8Training and Biostatistics Administration, Ministry of Health and Population, Alexandria 21561, Egypt; hiph.amiraelzorkany@alexu.edu.eg; 9Internal Medicine Department, Alexandria University, Alexandria 21526, Egypt; fatemaalabed94@gmail.com; 10General Surgery Department, Faculty of Medicine, Alexandria University, Alexandria 21561, Egypt; amr.alnagar@alexmed.edu.eg; 11University Hospital of Coventry and Warwickshire, Coventry CV2 2DX, UK; dr_nashwa.anwar@hotmail.com; 12Faculty of Communication and Media, King Abdulaziz University, Jeddah 22254, Saudi Arabia; malharthi5@kau.edu.sa; 13Biomedical Engineering Department, Collage of Engineering, Wraith Al-Anbiyaa University, Karbala 56001, Iraq; alikareem.mohsin@gmail.com; 14LabCom Research Unit, University of Beira Interior, Foundation for Science and Technology, 6201-001 Covilha, Portugal; bianca.toniolo@ubi.pt (B.T.); tamela.grafolin@ubi.pt (T.G.); 15Heidelberg Institute of Global Health, Heidelberg University, 69120 Heidelberg, Germany; thitthit.aye@uni-heidelberg.de; 16Department of Biomedical Informatics and Medical Statistics, Medical Research Institute, Alexandria University, Alexandria 21526, Egypt; drehsan.deghidy@yahoo.com; 17Pharmacy Department, Rompin Hospital, Ministry of Health Malaysia, 62590 Putrajaya, Malaysia; siti.sbahri@gmail.com; 18Mahidol-Oxford Tropical Medicine Research Unit, Faculty of Tropical Medicine, Mahidol University, Bangkok 10400, Thailand; jarntrah@tropmedres.ac; 19Department of Public Health, Medical Research Office, Sudanese Medical Research Association, Khartoum 11111, Sudan; hiph.yelhadi@alexu.edu.eg; 20Department of Women’s and Children’s Health, Uppsala University, 75236 Uppsala, Sweden; salmaelmukashfieltahir.mohammed.0806@student.uu.se; 21Department of Physiology, Faculty of Medicine, Al-Azhar University, Assiut 71524, Egypt; drnoor83@hotmail.com; 22Department of Neurology, Ibn Sina Hospital, Safat, Kuwait City 13115, Kuwait; dr.ismail.ibrahim2012@gmail.com; 23Institute of Public Health, College of Medicine and Health Sciences, United Arab Emirates University, Al Ain 15551, United Arab Emirates; ielbarazi@uaeu.ac.ae; 24College of Medicine, University of Sharjah, Sharjah 27272, United Arab Emirates; bsaddik@sharjah.ac.ae; 25Department of Global Public Health, Karolinska Institutet, 17176 Stockholm, Sweden; ziad.khatib@gmail.com; 26College of Pharmacy, Al-Zahraa University for Women, Karbala 56001, Iraq; hiba.akram@alzahraa.edu.iq; 27Hepatology Unit, Internal Medicine Department, Faculty of Medicine, Alexandria University, Alexandria 21131, Egypt; ahmed.kamal@alexmed.edu.eg

**Keywords:** COVID-19 vaccine, vaccine hesitancy, COVID-19, comment tone and position, content analysis, social media

## Abstract

Vaccine hesitancy (VH) is defined as a delayed in acceptance or refusal of vaccines despite availability of vaccination services. This multinational study examined user interaction with social media about COVID-19 vaccination. The study analyzed social media comments in 24 countries from five continents. In total, 5856 responses were analyzed; 83.5% of comments were from Facebook, while 16.5% were from Twitter. In Facebook, the overall vaccine acceptance was 40.3%; the lowest acceptance rates were evident in Jordan (8.5%), Oman (15.0%), Senegal (20.0%) and Morocco (20.7%) and the continental acceptance rate was the lowest in North America 22.6%. In Twitter, the overall acceptance rate was (41.5%); the lowest acceptance rate was found in Oman (14.3%), followed by USA (20.5%), and UK (23.3%) and the continental acceptance rate was the lowest in North America (20.5%), and Europe (29.7%). The differences in vaccine acceptance across countries and continents in Facebook and Twitter were statistically significant. Regarding the tone of the comments, in Facebook, countries that had the highest number of serious tone comments were Sweden (90.9%), USA (61.3%), and Thailand (58.8%). At continent level, serious comments were the highest in Asia (58.4%), followed by Africa (46.2%) and South America (46.2%). In Twitter, the highest serious tone was reported in Egypt (72.2%) while at continental level, the highest proportion of serious comments was observed in Asia (59.7%), followed by Europe (46.5%). The differences in tone across countries and continents in Facebook and Twitter and were statistically significant. There was a significant association between the tone and the position of comments. We concluded that the overall vaccine acceptance in social media was relatively low and varied across the studied countries and continents. Consequently, more in-depth studies are required to address causes of such VH and combat infodemics.

## 1. Introduction

The coronavirus disease 2019 (COVID-19) is caused by the severe acute respiratory syndrome coronavirus 2 (SARS-CoV-2). On 12 March 2020, the World Health Organization (WHO) declared COVID-19 as a pandemic [1]. This pandemic affected over 474.7 millions of people worldwide with over 6.1 million deaths [2]. The pattern of infection and mortality differed significantly across countries [3,4]. Healthcare workers and elderly people are at higher risk of acquiring the infection and related complications, but there is also an increase in the number of young persons who present with COVID-19 related complications [5]. These factors resulted in a high burden on healthcare facilities and the global economy in addition to the social drawbacks [6]. 

Indeed, there is a debate on the effectiveness of non-pharmaceutical interventions on viral transmission. Social distancing and facemasks failed to control the pandemic in Sweden [7], while the dependence on herd immunity strategy resulted in higher deaths [8]. On the other hand, many studies have proven the effectiveness of these measures on the pandemic containment [9,10,11]. Consequently, the need for effective vaccination has been urged, however, one of the major limiting elements for a wide coverage of vaccination programs, especially for newly developed vaccines, is vaccine hesitancy (VH) [12].

The WHO referred to VH as one of the ten greatest threats to global health in 2019 [13]. VH is defined as a delay in acceptance or refusal of vaccines despite availability of vaccination services [14]. The reasons for VH are numerous, complex across vaccines, countries, as well as time periods. Reasons include lack of confidence (i.e., belief in vaccine safety and effectiveness), complacency (i.e., not identifying the disease as high risk and vaccination as essential), constraints (i.e., practical limitations), collective responsibility (willingness to protect others by becoming vaccinated), and calculation (involvement in intensive information collection and a thorough analysis of the risks of diseases and vaccines) [15,16].

As of 16 April 2022, at least one dose of a COVID-19 vaccination has been administered to 65 % of the world’s population. Globally, 11.45 billion doses have been provided, with 11.79 million administered each day. Only 15.2% of low-income countries adults have gotten at least one dose [17]. Vaccination against COVID-19 has been effective in reducing mortality, progression to severe disease and human-to-human transmission [18,19], however, many concerns about the safety and efficacy of these vaccines have been raised. The protective effect of vaccination against hospitalization and mortality diminishes with time [20]. Moreover, vaccination has a reduced effectiveness against variant strains, for instance, when the SARS-CoV-2 Delta variant became common, the proportion of fully vaccinated patients infected with SARS-CoV-2 increased faster than expected [21]. Anti-vaxxers’ false theories spread through social media, including concerns about safety, confusion about protection levels, perceived risk and fears, poor health literacy, lack of awareness about the virus, misinformation or lack of accurate knowledge about the vaccines, concerns about safety in the elderly and people with various preexisting comorbidities, the fast-tracking of vaccines, doubts about the effectiveness of the available vaccines against variant strains, anti-vaccine myths, and confusing messages about some severe side effects of a few vaccines threatens the global efforts to control the circulation of SARS-CoV-2 [22,23]. The main concern is that unvaccinated individuals can act as reservoirs of SARS-CoV-2 and maintain the transmission cycle [23]. 

This “infodemic” (overabundance of health information, misinformation, and disinformation) may impede implementing the best public health policies which may be more critical and dangerous than the actual pandemic and may cost lives [24]. Rumors about COVID-19 vaccines were released on social media with thousands of likes, shares, retweets, and millions of views. Rumors ranged from talking about the non-existence of the COVID-19 pandemic, the non-existence of true vaccines, ineffective vaccination in certain races, being dangerous in elderly people, the ability of vaccines to alter human DNA or to control the behavior of persons, and up to the death of individuals after receiving vaccines [25]. In a large study the knowledge, attitude, and practices of 215,731 participants from 45 countries were assessed. The estimated overall correct answers for knowledge, good attitude and good practice were 75%, 74% and 70%, respectively [26]. Therefore, a serious collaborative initiative has started to remove the misleading claims to counteract the spread of misinformation about COVID-19 vaccines in social media platforms (i.e., Facebook, Twitter, and You Tube) [25]. In this study we aimed to determine opinions and attitudes toward COVID-19 vaccines through analyzing reactions and comments of social media users to the COVID-19 posts released by health authorities.

## 2. Materials and Methods

We conducted quantitative content analysis to analyze the attitude of social media users to COVID-19 vaccination [27]. We analyzed social media posts from official health institutions regarding COVID-19 vaccination and associated social media users’ comments for 24 countries. Researchers identified the main social media platforms, number of social media users, and the social media pages of the official health institutions in each country via Statista website [28]. Based on this information, and by considering the population of each country, the sample size of included posts and comments from each country was determined. For each country, the population number was determined and then after conducting a power analysis, sample size was determined in the form of the minimum required number of posts to be collected for this study. We determined the number of comments we should include from each social media platform based on the percentage of usage of each social media platform in each country [28].

Quantitative content analysis was used to fulfil the goal of this study. Content analysis is a research method to analyze verbal or visual communication/message [29]. There are two forms of content analysis: qualitative and quantitative. Quantitative content analysis is a branch of content analysis that analyzes messages by quantifying the occurrence of words, expressions, phrases, and so on [30]. The process of a quantitative content analysis begins with determining the different expressions to be extracted from the text. A coding process preceded this step, and a codebook was designed and used to quantify the occurrence of the words that provide inference to the needed expressions. Since this process has a human factor (i.e., during the coding process), multiple coders coded same content to ensure the reliability of the coding process. In this study, two waves of coding were carried out on the collected data.

A codebook was created as the main instrument used in this study. The codebook aimed to code social media users’ comments and reactions on COVID-19 vaccine posts identified in the previous step. In general, when selecting posts on social media, researchers analyzed this same post on the different social media platforms where it was posted. For Facebook posts, the data collected was screenshot/link to the post, date, time, location, source, language, number of comments, and number of reactions to the post.

### 2.1. Codebook

#### 2.1.1. Position of the Comment

“With vaccination” meant that the comment supported the vaccine’s existence and use, “Against vaccination” meant that the comment was refusing the vaccine in any means whether industry or intake, “Neutral” meant that the comment was simply a comment, not showing a directional attitude.

#### 2.1.2. Tone of the Comment

“Serious” meant that the comment was literal in its meaning, “Humorous” meant that the comment was funny “Sarcastic” when the comment has the character of sarcasm, “Opinion” meant that the comment was explaining the person’s point of view about the vaccine. 

To analyze the comments, the codebook included the following codes: post being analyzed, comment screenshot, tone of the comment (serious, humorous, opinion), opinion position of the comment (with vaccination, against vaccination, neutral), and the number of reactions to the comment. 

For Twitter, the following codes were included for each tweet: screenshot, date, time, location, source, language, number of replies, number of retweets, number of quote tweets, and number of likes. The following codes were used to analyze the replies: tweet ID, reply ID, screenshot, tone of the reply (serious, humorous, opinion), and comment position (with vaccination, against vaccination, neutral).

Before the start of the coding process, all the data collectors were trained on how to code comments. During the training, data collectors got a deep understanding of the concepts included in the codebook and the spectrum of meanings that can be included under each code. The training also involved sample coding activities of real comments. The training was carried out once all the data collectors had a mutual agreement when coding the sample comments. The tone and position codes were entered during the collection process from the interpretation of the researcher to the language of the comment and were assigned to one of three options of the tone and position by 0, 1 method.

The coding of the selected data was carried out by two teams of authors independently to run reliability testing. The first wave of coding was carried out starting on 15 January to 22 January. The second wave of coding was carried out by switching the researchers and coding the same comments for a second time. A screenshot/link of each comment was included in the codebook. A second coder coded the screenshot for the second time blindly.

### 2.2. Sampling Technique

Country selection: A convenient sampling technique was used to include the countries. 

The authors developed a selection strategy to include social media posts from the official pages of health authorities as noted below: 

In countries where the vaccine was delivered, the first post that addressed the delivery of the COVID-19 vaccine to the corresponding country was extracted and considered in the data collection process;

In countries where the vaccine had not been delivered yet, the first post that addressed the COVID-19 vaccine, in general, was extracted and considered in the data collection process. 

For example, if Nigeria had posts related to COVID-19 delivery in the country, the authors identified the social media pages of the official health channels in the country, then scrolled down to find the first post that talked about vaccine delivery, then started collecting the comments on this post, with the following post to be the one to be analyzed. If the authors checked the official pages and couldn’t find any posts related to vaccine delivery (because the vaccine had not yet been delivered to the country), the authors looked for any posts that talked about COVID-19 vaccine in general.

For Facebook, in each of the included posts, its comments were screened and collected through exploring the “most relevant” filter category in each port on Facebook. Comments were collected until reaching the determined sample size for each country. If the required number of comments wasn’t reached, the next post that addresses the COVID-19 vaccine was extracted and the comments were collected using the same method. For Twitter, systematic random sampling was applied by collecting every other reply to the desired tweet. We divided the number comments “N” by sample size “n” to calculate the sampling interval “i”. In case this value was in decimals, we rounded the figure to the nearest whole number/integer. Then, a random starting point, “r”, was chosen from where the sampling interval “i” is used to pick responses.

### 2.3. Statistical Analysis

All data were assessed using SPSS, version 26.0. Figures were built using ggplot2 package for R software version 4.1.2. Descriptive frequencies were calculated for qualitative variables (comments and tone). Categorical variables were presented in total numbers (n) and percentages of all recorded posts. The Chi-squared test was applied for evaluation of association between country and attitude or comments towards vaccination. Kappa statistic was performed to test for interrater agreement; (values < 0 as indicating no agreement and 0–0.20 as slight, 0.21–0.40 as fair, 0.41–0.60 as moderate, 0.61–0.80 as substantial, and 0.81–1 as almost perfect agreement). *p* values < 0.05 were considered significant. The Kappa value for the inter-coder reliability was 0.85, which indicated a strong agreement between the coders.

## 3. Results

### 3.1. Description of the Collected Data

In total, 4897 (83.5%) of comments were from Facebook, while 965 (16.5%) were from Twitter. The comments were written in English 46.0% (2269), Portuguese 15.6% (912), Arabic 20.5% (1203), German 6.6% (388), Malay 5.2% (307), Burmese 1.7% (100), Thai 1.5% (87), Spanish 1.8% (106), French 0.1(5) and Swedish 0.9% (55).

Comments were collected from 24 countries: United States of America (USA) (2176), Brazil (846), Saudi Arabia (385), United Kingdom (UK) (398), Egypt (381), Germany (388), Malaysia (307), Myanmar (100), Morocco (150), Mexico (106), Thailand (87), United Arab Emirates (UAE) (398), Tunisia (36), Portugal (66), Iraq (76), Sweden (55), Libya (20), Jordan (59), Palestine (45), Kuwait (28), Oman (27), Lebanon (13), Sudan (11), and Senegal (5). 

### 3.2. Position of Social Media Users towards COVID-19 Vaccine

#### 3.2.1. On the Country Level

##### Facebook

Acceptance rate: The overall vaccine acceptance in Facebook was 40.3% (1975/4897) in the countries. Countries that had a low acceptance rate were the USA (22%), UK (22.3%), Mexico (22.6%), and Palestine (24.4%), followed by Egypt (48.5%), Myanmar (57.0%), Thailand (50.6%), and Iraq (56.6%), Portugal (49.2%), and Germany (38.2%). Countries that had high acceptance rates were Saudi Arabia (88.3%), UAE (76.1%), Libya (75.0%), Brazil (67.5%), and Kuwait (65.2%). The difference in the acceptance rate among countries was statistically significant (*p* < 0.001) (Table 1). 

Rejection rate: UAE (3.3%), Myanmar (4.0%), Thailand (5.7%), Oman (5.0%) had the lowest rejection rates among all countries followed by that had a low rejection rate were Saudi Arabia (9.6%), Sweden (20.0%), Brazil (21.6%), Kuwait (21.7%), Mexico (20.8%), and Libya (25.0%). Countries that had medium rejection rates were Egypt (33.6%), Portugal (36.5%), Iraq (35.5%), Tunisia (38.9%), Malaysia (40.1%), Germany (52.9%), Morocco (53.3%), and Palestine (55.6%). Countries that had high rejection rates were Jordan (89.8%), UK (74.3%), USA (60.4%), Sudan (60.0%), and Senegal (60%). The difference in the rejection rate among countries was statistically significant (*p* < 0.001) (Table 1).

##### Twitter Acceptance

The highest acceptance rate was found in UAE (100.0%), followed by Lebanon (84.6%), and Brazil (81.1%). The lowest acceptance was detected in Sudan (0.0%), Oman (14.3%), UK (23.3%), and USA (20.5%).

#### 3.2.2. On the Continental Level

##### Facebook

Acceptance rate: The total acceptance rate was 40.3% (1975/4897); being lowest in North America with 22.6% (417/1844), Europe 35.8% (263/735), and Africa, 42.6% (249/584). The highest vaccine acceptance rate was in South America 67.5% (393/582) followed by Asia 56.7% (653/1152). The difference in the acceptance rate among the continents was statistically significant (*p* < 0.001) (Table 2).

Rejection Rate: The total rejection rate was 43.4% (2124/4897); the rate was the lowest in South America 21.6% (126/582). In Asia, the rejection rate was 24.2% (279/1152), while In Africa, the rejection rate was 39.4% (230/584). Rejection rate was the highest in North America was 58.1% (1071/1844). Followed by Europe 56.9% (418/735). The difference in the rejection rate among continents was statistically significant (*p* < 0.001) (Table 2).

##### Twitter

Acceptance rate: the overall acceptance rate was 41.5% (400/965); the highest acceptance rate was in South America (81.1%), and Asia (51.4%), followed by Africa (42.1%). The lowest acceptance was in North America (20.5%), and Europe (29.7%).

Rejection rate: the highest rejection rate was in North America (77.4%), and Europe (64.0%), followed by Africa (52.6%). South America witnessed the lowest rejection rate (15.2%) followed by Asia (22.2%).

### 3.3. Tone of the Social Media Comments towards COVID-19 Vaccine

#### 3.3.1. On the Country Level

##### Facebook

Countries that had the highest humorous comments were UAE (100%), Oman (95.0%), Libya (75.0%). Sarcastic comments were the highest in Sudan (20%), and Tunisia (16.7%). Countries that had the highest number of opinion comments were Senegal (40%), UK (29.7%), and Egypt (23.1%). Countries that had the highest number of serious comments were Sweden (90.9%), USA (61.3%), and Thailand (58.8%). The difference in the comments’ tone was statistically significant (*p* < 0.001) (Table 3).

##### Twitter

Brazil had the highest humorous tone of comments (13.3%). Germany had the highest sarcastic tone (35.0%). Oman, Portugal, Sudan, and UAE had the highest number of opinion- tone (100%) for each. The highest serious tone was reported in Egypt (72.2%).

#### 3.3.2. On the Continental Level

Facebook: Serious comments were the highest in Asia 58.4% (673/1152), Africa 46.2% (270/584), South America 46.2% (269/582), followed by Europe 33.7% (248/735), and North America 25.3% (467/1844). Humorous comments were the greatest in Africa 43/584 (7.4%), Europe 32/735 (4.4%), followed by Asia 47/1152 (7.2%), South America 23/582 (4.0), and North America 45/1884 (2.4%). Sarcastic comments were 22.0% (162/735) of comments from Europe, 20.7% (121/584) of comments from Africa, 12.5% (231/1844) of comments from North America, 6.2% (71/1152) of comments from Asia, and 5.3% (31/582) of comments from South America. Opinion comments were highest in North America 59.7% (1101/1844), then South America 44.5% (259/582), Europe 39.9% (5293/735), Asia 31.3% (361/1152), and finally Africa 25.7% (150/584). The difference between the continents regarding the tone of the comments was statistically significant (*p* < 0.001) (Table 4). 

Twitter: the highest proportion of serious comments was observed in Asia 59.7% (42/72), followed by Europe 46.5% (80/172). South America had the highest proportion of humorous comments 13.4% (35/264) followed by Europe 4.7 (8/172). Europe and North America had the highest sarcastic comments (30.2% and 21.5%, respectively). Africa reported the highest proportion of opinion comments 68.4% (13/19) followed by North America 51.1% (224/438) (Table 4).

Table 5 showed that there was a significant correlation between tone of the comment and comment position.

## 4. Discussion

Social media is an essential tool that enables healthcare specialists to share information, connect with the public, and interact with patients, students, and colleagues [31]. It also permits individuals to share information and ideas. Many organizations use social media to disclose important updates on different situations. While it provides extraordinary capability for the public to communicate, social media has also been a major factor in the ascent of sentiments and opinions that are damaging to public health, especially amid the COVID-19 pandemic. Rapid and wide vaccination all over the world is mandatory to control the pandemic, support reopening plans, save healthcare systems and economies, and avoid a disconnect between countries [32]. However, rumors and misinformation about COVID-19 vaccines have spread widely over social media platforms. Globally, this spread of misinformation has greatly affected population attitudes towards vaccination and has increased VH among social media users worldwide so that understanding VH via the perspective of social media is critical [33,34].

This study analyzed 5862 social media comments in 24 countries from five continents around the world, with ten different languages included in this analysis. The study described the tone of the comments (serious, humorous, sarcastic, or opinion) in these countries as well as the position of the comments (with vaccination, against vaccination, or neutral) towards COVID-19 vaccination. There was a significant difference between countries and continents regarding the tone and the position of the comments. Moreover, there was a significant association between the tone and the position of the comment.

Overall rejection of COVID-19 vaccines among social media users’ comments was 41.3% (43.4% in Facebook 77.4% in Twitter). The highest rejection was among social media users from North America. Rejection rates exceeded 60% of comments from Jordan, USA, UK, and Sudan, while it was less than 10% among comments from UAE, Myanmar, Thailand, Lebanon, and Oman all of which are Asian countries. Acceptance rates were more than 50% among comments from Lebanon, Saudi Arabia, UAE, Libya, Brazil, Kuwait, and Portugal. More than 50% of comments from Oman and Mexico were neutral. It is well established that COVID-19 vaccine acceptance and rejection across countries and continents may be attributed to different factors including trust in health authority, level of country income, vaccine availability, trust in vaccine effectiveness and side effects, and conspiracy theory [35,36]. The same point was addressed by Nuzhath et al. [37] who recently published a study on COVID-19 tweets posted in English during November 2020. They found that tweets against COVID-19 vaccination were greater than positive tweets. Griffith et al. [38] addressed causes of VH among tweets from Canada. The social media users’ concerns were mainly about safety, importance, mistrust towards the medical industry, and suspicion about economic forces. The phenomenon of high rejection rates among social media users is an alarming sign. As anti-vaccine propaganda is widely disseminated on social media, early research has shown that exposure to such information may directly alter vaccination beliefs and lead to downstream VH [39,40,41]. In fact, certain users, such as those with cognitive disability, older age, lower literacy, and less digital literacy, have been shown to be more sensitive to these narrative emotional appeals on social media [42]. Prior to being exposed to social media information, users’ basic personal values and prejudices, such as ethno-cultural, religious, or political convictions, may affect their response to such messages [43]. For instance, the pandemic and COVID-19 vaccination have impacted people’s religious behaviors as attending in religious places with crowds and gatherings. The application of vaccine mandates decreased the attendance in religious places, with lack of trust in the experts and stories of human rights’ violations, that negatively affected people’s attitudes towards the vaccine [44].

It is worthy to note that coverage of COVID-19 vaccination is still low in many countries with high acceptance rates to the vaccines among social media users. However, many countries that have shown higher refusal rates have reached high rates of vaccination coverage. This may be due to vaccine inequity, in which there is an unequal distribution of COVID-19 vaccine availability internationally [24]. This inequity in vaccine distribution can endanger the world and can compromise vaccines effectiveness due to the increased risk for the development of different variants [25]. 

We speculate that most of the comments against vaccination were because of safety issues, questions on long-term complications, mistrust with global health organizations, disbeliefs, and rumors that had already been proliferated throughout different social media platforms, such as the belief that COVID-19 is a mild or even a non-existent disease. As a result, it is critical that all stakeholders participating in the COVID-19 immunization program recognize the detrimental impact of infodemic and disinformation on these efforts and actively work to counteract them [34,45]. In this case, key stakeholders include research scientists involved in vaccine development, pharmaceutical companies manufacturing these vaccines, health care professionals administering vaccines, public health experts, ministries, and departments of health in charge of funding and monitoring vaccination programs, electronic and print media, and the community itself. To make this worldwide immunization program a success, they must all coordinate and collaborate [46].

Currently, Facebook users may utilize seven reactions (‘like’, ‘love’, ‘care’, ‘wow’, ‘angry’, ‘sad’, ‘ha-ha’) to respond to postings using emoticons. A vast number of possible replies is a relatively recent phenomena, first appearing on Facebook in 2016. So far, there haven’t been many articles that investigate how people interact with these dynamic replies. In this study emotions were used to assess tone of comments toward vaccination. We linked the tone of the comment with the position toward vaccination. The comments with more serious tones lean towards vaccine acceptance while the opinionated comments lean towards vaccine rejection. On the other hand, Wawrzuta et al. [47] analyzed Facebook comments related to the different events related to COVID-19 vaccination; (vaccine introduction, announcements of vaccine efficacy, vaccines registration, and the first vaccination in Poland). They noticed that, while the comments were mostly negative, the reactions were positive. This highlights the importance of spreading the scientific facts about the danger of COVID-19 on persons and communities, as well as prioritizing misconceptions about COVID-19 vaccines to increase COVID-19 vaccination acceptance [45].

Based on the total number of data collected from Facebook and twitter, the most frequent tone was opinion, (2164/4897 and 317/965, respectively). Similarly, Lyu et al. [48] found that opinion was the most tweeted issue and remained a hotly debated topic for the length of the study. This may reflect the positive attitude of the population to combat the spread of the pandemic and their desire to return to normal life.

### Strength and Limitations

To the best of our knowledge, this study is the first to address the COVID-19 VH worldwide through content analysis. Researchers included different countries with variable cultures and different socioeconomic statuses. Another strength of our study was that we collected data from two different social media platforms (Twitter and Facebook). The main limitation was that some a big country such as China was not included in the study as the Chinese do not use social media such as Facebook and Twitter. Another limitation was that countries were selected based on data collector’s availability, which was important to ensure that data collectors were familiar with the cultural context of the studied countries so that we cannot generalize our findings on global level. Finally, the opinion of social network users does not necessarily reflect the opinion of people in a country, and that user may vary greatly in their characteristics from one country to another, with which the results would not be directly comparable.

## 5. Conclusions

Qualitative analysis could be an important avenue for future work to explore in order to acquire more insight on the themes that shape users’ interaction about COVID-19 vaccination and identify infodemics Analysis of comments on COVID-19 vaccination in different social media revealed significant differences of respondents’ attitude toward vaccination. The overall vaccine acceptance in social media is relatively low and varied across the studied countries and continents. Moreover, the tone of the comments on social media reflected the position towards COVID-19 vaccination. Health authorities should continuously provide health education and information to citizen through social media channels. In fact, our anti-vaccine codebook can assist public health professionals in better understanding social media content. Early identification of societal doubts enables focused communication initiatives. Furthermore, our codebook can be used in future analyses by researchers and public health professionals who are monitoring society’s reaction to the COVID-19 vaccination. Consequently, more in-depth studies are required to address causes of such VH and combat infodemics that may affect the position toward vaccination.

## Figures and Tables

**Table 1 ijerph-19-05737-t001:** Country-position toward COVID-19 vaccination in Facebook and twitter.

Facebook (*n* = 4897)		With Vaccination (*n* = 1975)	Against Vaccination (*n* = 2124)	Neutral (*n* = 798)	*p* *
**Country**	Brazil	393 (67.5)	126 (21.6)	63 (10.8)	<0.001
	Egypt	176 (48.5)	122 (33.6)	65 (17.9)	
	Germany	133 (38.2)	184 (52.9)	31 (8.9)	
	Iraq	43 (56.6)	27 (35.5)	6 (7.9)	
	Jordan	5 (8.5)	53 (89.8)	1 (1.7)	
	Kuwait	15 (65.2)	5 (21.7)	3 (13.0)	
	Libya	15 (75.0)	5 (25.0)	0 (0.0)	
	Malaysia	102 (33.2)	123 (40.1)	82 (26.7)	
	Mexico	24 (22.6)	22 (20.8)	60 (56.6)	
	Myanmar	57 (57.0)	4 (4.0)	39 (39.0)	
	Oman	3 (15.0)	1 (5.0)	16 (80.0)	
	Palestine	11 (24.4)	25 (55.6)	9 (20.0)	
	Portugal	31 (49.2)	23 (36.5)	9 (14.3)	
	Saudi Arabia	303 (88.3)	33 (9.6)	7 (2.0)	
	Senegal	1 (20.0)	3 (60.0)	1 (20.0)	
	Sudan	4 (40.0)	6 (60.0)	0 (0.0.)	
	Sweden	40 (72.7)	11 (20.0)	4 (7.3)	
	Thailand	44 (50.6)	5 (5.7)	38 (43.7)	
	Tunisia	22 (61.1)	14 (38.9)	0 (0.0)	
	UAE	70 (76.1)	3 (3.3)	19 (20.7)	
	UK	59 (21.9)	200 (74.3)	10 (3.7)	
	USA	393 (22.6)	1049 (60.4)	296 (17.0)	
	Morocco	31 (20.7)	80 (53.3)	39 (26.0)	
**Twitter** **(n = 965)**		**With vaccination** **(*n* = 400)**	**Against vaccination** **(*n* = 515)**	**Neutral** **(*n* = 50)**	** *p* **
**Country**	Brazil	214 (81.1)	40 (15.2)	10 (3.8)	<0.001
	Egypt	8 (44.4)	9 (50.0)	1 (5.6)	
	Germany	19 (47.5)	17 (42.5)	4 (10.0)	
	Kuwait	4 (80.0)	1 (20.0)	0 (0.0)	
	Lebanon	11 (84.6)	1 (7.7)	1 (7.7)	
	Oman	1 (14.3)	1 (14.3)	5 (71.4)	
	Portugal	2 (66.7)	0 (0.0)	1 (33.3)	
	Saudi Arabia	16 (38.1)	13 (31.0)	13 (31.0)	
	Sudan	0 (0.0)	1 (100.0)	0 (0.0)	
	UAE *	5 (100.0)	0 (0.0)	0 (0.0)	
	UK *	30 (23.3)	93 (72.1)	6 (4.7)	
	USA *	90 (20.5)	339 (77.4)	9 (2.1)	

* Test statistics Chi Square, χ^2^ = 1472, df = 44, Test statistics: Monte Carlo (χ^2^ = 446, df = 22), * UAE: United arab of emirates; UK: United Kingdom; USA: United States of America.

**Table 2 ijerph-19-05737-t002:** Continent-position toward COVID-19 vaccination.

**Facebook**			**Comment Position**			** *p* **
		**Total** **(*n* = 4897)**	**With Vaccination** **(*n* = 1975)**	**Against Vaccination** **(*n* = 2124)**	**Neutral** **(*n* = 798)**	
Continents	Africa	584	249 (42.6)	230 (39.4)	105 (18.0)	<0.001
	Asia	1152	653 (56.7)	279 (24.2)	220 (19.1)	
	Europe	735	263 (35.8)	418 (56.9)	54 (7.3)	
	North America	1844	417 (22.6)	1071 (58.1)	356 (19.3)	
	South America	582	393 (67.5)	126 (21.6)	63 (10.8)	
**Twitter**		**Total** **(*n* = 965)**	**With Vaccination** **(*n* = 400)**	**Against Vaccination** **(*n* = 515)**	**Neutral** **(*n* = 50)**	** *p* **
Continents	Africa	19	8 (42.1)	10 (52.6)	1 (5.3)	<0.001
	Asia	72	37 (51.4)	16 (22.2)	19 (26.4)	
	Europe	172	51 (29.7)	110 (64.0)	11 (6.4)	
	North America	438	90 (20.5)	339 (77.4)	9 (2.1)	
	South America	264	214 (81.1)	40 (15.2)	10 (3.8)	

Test statistics: Chi Square, df = 8, χ^2^ = 680; Test statistics: Chi square (χ^2^ = 295, df = 12).

**Table 3 ijerph-19-05737-t003:** Country-Tone in Facebook.

**Facebook** **(*n* = 4897)**		**Humorous** **(*n* = 1927)**	**Sarcastic** **(*n* = 190)**	**Opinion** **(*n* = 616)**	**Serious** **(*n* = 2164)**	** *p* **
**Country**	Brazil	269 (46.2)	23 (4.0)	31 (5.3)	259 (44.5)	<0.001
	Egypt	180 (49.6)	28 (7.7)	84 (23.1)	71 (19.6)	
	Germany	96 (27.60)	13 (3.7)	73 (21.0)	166 (47.7)	
	Iraq	53 (69.7)	9 (11.8)	14 (18.4)	0 (0.0)	
	Jordan	19 (32.2)	4 (6.8)	22 (37.3)	14 (23.7)	
	Kuwait	14)60.9)	0 (0.0)	1 (4.3)	8 (34.8)	
	Libya	15 (75.0)	0 (0.0)	4 (20.0)	1 (5.0)	
	Malaysia	164 (53.4)	20 (6.5)	12 (3.9)	111 (36.2)	
	Mexico	38 (35.8)	7 (6.6)	26 (25.4)	35 (33.0)	
	Myanmar	50 (50.0)	4 (4.0)	8 (8.0)	38 (38.0)	
	Oman	19 (95.0)	0 (0.0)	0 (0.0)	1 (5.0)	
	Palestine	16 (35.6)	1 (2.2)	4 (8.9)	24 (53.3)	
	Portugal	43 (69.3)	1 (1.6)	9 (14.3)	10 (15.9)	
	Saudi Arabia	215 (62.7)	5 (1.5)	9 (2.6)	114 (33.2)	
	Senegal	1 (20.0)	0 (0.0)	2 (40.0)	2 (40.0)	
	Sudan	5 (50.0)	2 (20.0)	3 (30.0)	0 (0.0)	
	Sweden	4 (7.3)	1 (1.8)	0 (0.0)	50 (90.9)	
	Thailand	31 (35.6)	4 (4.6)	1 (1.1)	51 (58.6)	
	Tunisia	22 (61.1)	6 (16.7)	6 (16.7)	2 (5.6)	
	UAE	92 (100.0)	0 (0.0)	0 (0.0)	0 (0.0)	
	UK	105 (39.0)	17 (6.3)	80 (29.7)	67 (24.9)	
	USA	429 (24.7)	38 (2.2)	205 (11.8)	1066 (61.3)	
	Morocco	47 (31.3)	7 (4.7)	22 (14.7)	74 (49.3)	
**Twitter**		**Humorous** **(*n* = 46)**	**Sarcastic** **(*n* = 167)**	**Opinion** **(*n* = 317)**	**Serious** **(*n* = 435)**	** *p* **
**Country**	Brazil	35 (13.3)	11 (4.2)	191 (72.3)	27 (10.2)	<0.001
	Egypt	0 (0.0)	2 (11.1)	3 (16.7)	13 (72.2)	
	Germany	0 (0.0)	14 (35.0)	9 (22.5)	17 (42.5)	
	Kuwait	0 (0.0)	1 (20.0)	3 (60.0)	1 (20.0)	
	Lebanon	0 (0.0)	1 (7.7)	11 (84.6)	1 (7.7)	
	Oman	0 (0.0)	0 (0.0)	7 (100.0)	0 (0.0)	
	Portugal	0 (0.0)	0 (0.0)	3 (100.0)	0 (0.0)	
	Saudi Arabia	0 (0.0)	6 (14.3)	17 (40.5)	19 (45.2)	
	Sudan	0 (0.0)	0 (0.0)	1 (100.0)	0 (0.0)	
	UAE *	0 (0.0)	0 (0.0)	5 (100.0)	0 (0.0)	
	UK *	8 (6.2)	38 (29.5)	68 (52.7)	15 (11.6)	
	USA *	3 (0.7)	94 (21.5)	117 (26.7)	224 (51.1)	

Test statistics: Monte Carlo (χ^2^ = 1073, df = 66), Test statistics: Monte Carlo (χ^2^ = 339, df = 33), * UAE: United arab of emirates; UK: United Kingdom; USA: United States of America.

**Table 4 ijerph-19-05737-t004:** Continents tone to COVID-19 vaccination in Facebook and Twitter.

**Facebook**		**Total** **(*n* = 4897)**	**Serious** **(*n* = 1927)**	**Humorous** **(*n* = 190)**	**Sarcastic** **(*n* = 616)**	**Opinion** **(*n* = 2164)**	***p* ***
Continents	Africa	584	270 (46.2)	43 (7.4)	121 (20.7)	150 (25.7)	<0.001
	Asia	1152	673 (58.4)	47 (4.1)	71 (6.2)	361 (31.3)	
	Europe	735	248 (33.7)	32 (4.4)	162 (22.0)	293 (39.9)	
	North America	1844	467 (25.3)	45 (2.4)	231 (12.5)	1101 (59.7)	
	South America	582	269 (46.2)	23 (4.0)	31 (5.3)	259 (44.5)	
**Twitter**		**Comment Tone**					** *p* **
		**Total** **(*n* = 965)**	**Serious** **(*n* = 435)**	**Humorous** **(*n* = 46)**	**Sarcastic** **(*n* = 167)**	**Opinion** **(*n* = 317)**	
Continents	Africa	19	4 (21.1)	0 (0.0)	2 (10.5)	13 (68.4)	*p* < 0.001
	Asia	72	43 (59.7)	0 (0.0)	8 (11.1)	21 (29.2)	
	Europe	172	80 (46.5)	8 (4.7)	52 (30.2)	32 (18.6)	
	North America	438	117 (26.7)	3 (0.7)	94 (21.5)	224 (51.1)	
	South America	264	191 (72.3)	35 (13.3)	11 (4.2)	27 (10.2)	

Test statistics: Chi Square (χ^2^ = 584, df = 12), Test statistics: Chi square (χ^2^ = 295, df = 12).

**Table 5 ijerph-19-05737-t005:** Overall relation between comment tone and comment position.

		Comment Tone				*p*
		Serious	Humorous	Sarcastic	Opinion	
**Comment Position**	With	1275 (53.7)	76 (3.2)	97 (4.1)	927 (39.0)	*p* < 0.001
	Against	732 (27.7)	119 (4.5)	592 (22.4)	1196 (45.3	
	Neutral	355 (41.9)	41 (4.8)	94 (11.1)	358 (42.2)	

Test statistics: Chi square (χ^2^ = 547, df = 6).

## Data Availability

The data presented in this study are available on request from the corresponding author.

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
