# Peer review of "COVID-19 Vaccine Acceptance among Social Media Users: A Content Analysis, Multi-Continent Study"

_ijerph, 2022, doi:10.3390/ijerph19095737_

Round 1

Reviewer 1 Report

This article analyses a selection of comments about the Covid19 pandemic virus vaccines on two specific social networks in a number of countries located on several continents.

I will comment on each of the sections below.
Regarding the authors, I am struck by the fact that 27 authors are included. I understand that this is a multi-country study, but I think it would be better to indicate the main coordinators of the study as authors, and mention the rest as part of the research group.

Summary:
Should be revised according to changes in the document after revision of each section.

Introduction:
The introduction summarises what the covid19 pandemic is and its impact on people's health. The role of vaccines in this pandemic is indicated. It also introduces the concept of vaccine hesitance, and focuses on the context of the current pandemic. Finally, the objective of the research is stated: to determine reactions to SARS-CoV-2 vaccines among social network users.
As a point for improvement, I think it is not correct to talk about the need for a high percentage of vaccinated population to achieve herd immunity (paragraph included in quote number 7), since the vaccines currently used are not sterilising. They have an effect on the reduction of infections (although less and less with the new variants) and above all on the reduction of hospitalisations, ICU admissions and mortality, but they totally prevent human-to-human transmission. Therefore, a high percentage of vaccinated population is necessary to reduce the burden on the health care system and to reduce mortality in the population, but not to achieve herd immunity.

Materials and methods:
It is explained that content analysis was used as the research methodology. It explains how the sample size was determined, the social networks where the comments were analysed (Facebook and Twitter), the random data collection strategy, the analysis of the comments, as well as the triangulation strategy between researchers to assess concordance. 
As an area for improvement, I believe that the most subjective part of the research is the classification of comments according to tone (serious, humorous and opinion) and position in relation to vaccines (praise, criticism, neutral). It would be useful to elaborate on what criteria were agreed upon by the researchers to include each comment in these categories.
On the other hand, the position and tone are from the comments of users of two very specific social networks (Facebook and Twitter). Perhaps it would be necessary to describe what characteristics (socio-demographic, etc.) the users of these two social networks have and whether they are the same in all countries and geographical areas or whether they vary.
Finally, it would be necessary to justify whether a random selection of comments on social networks guarantees a true representation of the comments, given that the tone and position are not distributed in my opinion randomly, but follow patterns more typical of chaotic systems, with comments that act as attractors and cause chains of comments for and against, which may then be followed by other more neutral comments for some time.

Results:
General data on the number of comments on each social network, language and countries are described.
Results are described according to position in relation to vaccines, grouped by continent and then by country. Statistical test results are presented analysing the relationship between position and continent and country. 
The same is then done for the tone of the comments. 
Finally, the relationship between position in relation to vaccines and tone of comments is analysed. 
As a point of improvement, I think that since it is not a random sample of countries within each continent, but a convenience sample, the results should not be analysed at the continent level, but only at the country level. In my opinion, it is not the same to say that in Africa the position and tone is X, as it is to say that in the countries analysed, which are located in Africa, the position and tone is X. Perhaps one could group the countries by continent, but not make analyses by saying that these are the results of a continent, because that is not real.

Discussion:
Starts the discussion by talking about the role of social networks in the transmission of information and disinformation. It then summarises the main findings on the position of comments on vaccines and compares them with the few existing studies. 
This is followed by a commentary on the inequity of vaccine distribution.
Then, the reasons for negative comments on vaccines are mentioned. 
This is followed by a very brief mention of results on the tone of the comments.
Finally, a section on limitations and strengths of the study is presented.
As areas for improvement, I believe that the comments about inequity, while true, are not issues that derive from the research results. The favourable or unfavourable opinion of social network users does not necessarily reflect the opinion of the country's population. It is not justified by the results that in all countries low vaccination rates and favourable opinion on vaccines are caused by a lack of vaccines. If this is considered to be the case, it would have to be argued with more data, and knowing that this favourable opinion is only that of social network users.
On the other hand, the results do not include details of the reasons given in the comments. If we want to discuss this data, it should appear in the results section.
Given that there are results on position, tone and the relationship between the two, I think that the discussion lacks more analysis on the results of the tone of the comments, and the whole analysis of the relationship between the two is missing.
Finally, in the limitations section, more weaknesses should be included, such as that the opinion of social network users does not necessarily reflect the opinion of people in a country, and that users may vary greatly in their characteristics from one country to another, with which the results would not be directly comparable.

Conclusions: 
The main findings, which answer the research objective, are summarised.

Reviewer 2 Report

Abstract:

  • Lines 61-62: What is meant by tone and position of the comments? And how does these relate to vaccine hesitancy?

Introduction:

  • Lines 78-80: Limited adherence to social distancing, mask mandates, and herd immunity resulted in deaths worldwide. Since this paper is described as multi-continent content analysis, I recommend incorporating cites that reflect this trend internationally, are not those limited to Sweden.
  • Line 102: Phrase that begins with “Therefore, a serious collaborative initiative…” is a fragment and not a full sentence.
  • Line 108: what is meant by “the levels of COVID-19 vaccines' reactions on social media?” Please clarify and explain in the paper.
  • Lines 111-112: We conducted quantitative content analysis to analyze the responses of social media users to COVID-19 vaccination – should this be user responses to COVID-19 vaccination information?

Methods:

  • The strategy for selecting social media content is unclear. I understand that the population of each country was considered and used to determine the power analysis and that the social media posts from the official pages of health authorities were also included. It is also my understanding that the authors collected data (comments) from the first post about vaccines on those pages were collected and analyzed. How many posts were collected in all (e.g., 1 per country?). Please include this information. Also, of the above interpretation is correct, I recommend cutting down this entire explanation and only including relevant, step by step information about the data collection process.
  • Missing a full codebook with descriptions of categories and examples used to complete the analysis. This is crucial for replication purposes and to ensure the data are credible and reliable. I do see that the authors included some of this information in the text of the paper (e.g., lines 164-165: tone of the comment (serious, humorous, opinion), opinion position of the comment (praise, criticism, neutral), and the number of reactions to the comment). However, the paper is missing the categories, descriptions, and examples of each theme (e.g., tone) and sub-theme (e.g., criticism) used to code the data. I recommend presenting this information as a Table in the paper.
  • Line 168-170: The authors explain that reactions to replies were used to analyze the tweets. How were reactions to replies included in the analysis of the Twitter data? Please explain and justify.
  • Thank you for explaining the inter-coder reliability testing using Cohen kappa. Please include the ranges for intercoder reliability across the data categories.

Results:

  • What was the rationale for combining the Facebook and Twitter data in the analysis? Please explain and justify. 
  • How were acceptance rates and rejection rates conceptualized and operationalized in the data collection and analysis? Please also provide this information for those data coded as neutral.
  • It is unclear what with, against, and neutral are describing in the category labels in the paper and in Table 1. In other words, how do the codes, “with, against, and neutral” relate – if at all - to tone, vaccine hesitancy (accept, reject?). Please explain in the paper and leave as a note in the table.
  • Chi square tests and relevant data (overall value of the chi-square, df, p values, etc.) are not included for data presented in Tables 1 and 2. Please include this information in the paper.
  • The paper is missing an in-text explanation of the chi square tests and relevant data (overall value of the chi-square, df, p values, etc.) for the findings presented Table 3. Without this information, there is little the reader can do to accurately interpret the findings.

Reviewer 3 Report

Ever since the COVID-19 vaccines was authorized a heated debate over its usefulness and safety emerged. Thus, the unquestionable strength of the manuscript is that its topic is very important and timely and the research itself was designed and described clearly. Another advantage of this research is that while most previous studies focused on the vaccine hesitancy (VH) among healthcare professionals or lay persons, still there is a scarcity of work on the VH in social media. Moreover, as the research covers material from 24 countries it can help to understand the importance of socio-cultural context of decision making. However, while it could be of interest to the readers of the Journal I believe that there are some key issues that that should be elaborated more deeply.

1. Although the abstract is easy to read, your contribution must be articulated more strongly there, i.e. your results are more complex and should be articulated at greater length. For example, highlight the interesting differences among countries.

2. While in the ‘Introduction’ the Authors describe key issues related to the vaccine hesitancy some more information on the public’s knowledge and attitudes towards the COVID-19 vaccine could be provided, i.e. due to novelty of this biotechnology more information about the pros and cons of the COVID-19 vaccine should be described. Especially, that there are several various vaccines which provoke different opinions.

Finally, the public’s vaccine hesitancy should not be reduced to “anti-vaxxers’ false theories” as there are also many cultural factors behind it. For example, for some communities it resulted from moral concerns related to the fact that some vaccine manufacturers used abortion-derived fetal cell lines:

-- Zimmerman RK. Helping patients with ethical concerns about COVID-19 vaccines in light of fetal cell lines used in some COVID-19 vaccines. Vaccine. 2021;39(31):4242-4244. doi:10.1016/j.vaccine.2021.06.027

-- Garcia LL, Yap JFC. The role of religiosity in COVID-19 vaccine hesitancy. J Public Health (Oxf). 2021 Sep 22;43(3):e529-e530. doi: 10.1093/pubmed/fdab192.

3. The Authors could briefly explain the concept of ‘infodemic’ in the reference to the COVID-19 pandemic, i.e.

-- Farooq F, Rathore FA. COVID-19 Vaccination and the Challenge of Infodemic and Disinformation. J Korean Med Sci. 2021;36(10):e78. doi: 10.3346/jkms.2021.36.e78.

-- Wilson SL, Wiysonge C. Social media and vaccine hesitancy. BMJ Glob Health. 2020;5(10):e004206. doi: 10.1136/bmjgh-2020-004206. 

-- Puri N, Coomes EA, Haghbayan H, Gunaratne K. Social media and vaccine hesitancy: new updates for the era of COVID-19 and globalized infectious diseases. Hum Vaccin Immunother. 2020;16(11):2586-2593. doi: 10.1080/21645515.2020.1780846. 

-- Ghaddar A, Khandaqji S, Awad Z, Kansoun R. Conspiracy beliefs and vaccination intent for COVID-19 in an infodemic. PLoS One. 2022;17(1):e0261559. doi: 10.1371/journal.pone.0261559.

4. Please clarify the criteria for selecting countries involved in the study.

5. Although the Authors are aware that one possible limitation of the study is that it focused solely on quantitative analysis I wonder whether they could provide at least a small sample of posts or tweets in favor, against and neutral or provide more detailed information on its coding and explain the criteria for its classification.

6. What was the rate of agreement between the coders and how did you resolve the possible discrepancies?

7. Tables: I am not sure whether what does ‘with’ mean, i.e. shouldn’t it be ‘pros’ or ‘in favour’?

8. Although both the ‘Method’ and ‘Results’ sections are presented clearly the paper lacks requires at least some description and explanation of the differences between various countries. Indeed, the ‘Discussion’ section fails to describe contextual differences between the countries: neither do we know the differences between morbidity and mortality rates, health policy during the pandemic, including restrictions, and vaccination campaigns or vaccination rates in each county (possible Table?). Consequently, without such information the readers can wonder what was the purpose of collecting the data from so many countries. Additionally, the results are purely descriptive and make it difficult to interpret the results.

9. There are at least couple other research on the impact of social media on the COVID-19 VH that could be addressed in the Discussion section, i.e.:

-- Muric G, Wu Y, Ferrara E. COVID-19 Vaccine Hesitancy on Social Media: Building a Public Twitter Data Set of Antivaccine Content, Vaccine Misinformation, and Conspiracies. JMIR Public Health Surveill. 2021 Nov 17;7(11):e30642. doi: 10.2196/30642.

-- Wilson SL, Wiysonge C. Social media and vaccine hesitancy. BMJ Global Health 2020;5:e004206

-- Tang L, Douglas S, Laila A. Among sheeples and antivaxxers: Social media responses to COVID-19 vaccine news posted by Canadian news organizations, and recommendations to counter vaccine hesitancy. Can Commun Dis Rep 2021;47(12):524–33. https://doi.org/10.14745/ccdr.v47i12a03

-- Gisondi MA, Barber R, Faust JS, Raja A, Strehlow MC, Westafer LM, Gottlieb M. A Deadly Infodemic: Social Media and the Power of COVID-19 Misinformation. J Med Internet Res 2022;24(2):e35552. doi: 10.2196/35552

-- Lyu JC, Han EL, Luli GK. COVID-19 Vaccine-Related Discussion on Twitter: Topic Modeling and Sentiment Analysis. J Med Internet Res. 2021;23(6):e24435. doi: 10.2196/24435.

-- Eibensteiner F, Ritschl V, Nawaz FA, Fazel SS, Tsagkaris C, Kulnik ST, Crutzen R, Klager E, Völkl-Kernstock S, Schaden E, Kletecka-Pulker M, Willschke H, Atanasov AG. People's Willingness to Vaccinate Against COVID-19 Despite Their Safety Concerns: Twitter Poll Analysis. J Med Internet Res. 2021;23(4):e28973. doi: 10.2196/28973. 

10. ‘Study limitations’ should be separated from the ‘Conclusions’ section.

11. Finally, the Conclusion section itself should be more reflective of policy implications of the findings. For example, the Authors could reflect on what systemic approaches should be undertaken to fight the ‘infodemic’ or promote credible information on vaccination in social media.

To conclude, while the issues raised in the paper are important and timely I believe that before it could be reconsidered for publication in the Journal it needs some substantial revision of theoretical parts of the paper. Also the literature on the impact of social media on VH needs to be updated. Most importantly, the Authors should explain the differences between the countries.

Round 2

Reviewer 1 Report

Thank you for the modifications made to the manuscript. They respond to comments made by reviewers and I think they improve the quality of the article. In my opinion, the discussion could still be broadened and improved a little further, but I have no problem about the article being admitted as it stands now. It would be necessary to review some sentences (repetition in the section of limitations) and spelling (missing capital letters in a sentence of the conclusions).

Author Response

Dear Reviewer,

 Thank you for your comments.

We addressed your comments by adding more info to the discussion with recent article citation.

The limitations and conclusion are now fixed.

Regards,

Reviewer 3 Report

The Authors have clarified all issues raised in the review and I believe that this revised manuscript is now more consistent owing to their corrections and additional arguments. On the whole, I appreciate this effort and have no further concern regarding the manuscript except from some minor remarks regarding the Abstract which contains some errors, i.e., line 63: the lowest/the highest: “the lowest acceptance rates were the highest in Oman (15.0%), Senegal (20.0%), Morocco (20.7%), and Jordan (8.5%)”. Additionally, it is recommended to list those countries from the lowest acceptance: Jorda, Oman, Senegal, Marocco. Also, line 68: “The differences in vaccine acceptance across countries and continents in Facebook and Twitter and were statistically significant”. Moreover, from reading abstract alone the readers can wonder what does ‘serious comments’ meant. Please rephrase. Finally, while referring to religious factors affecting VH the Authors might check recently published paper: Jones, D.G. Religious Concerns About COVID-19 Vaccines: From Abortion to Religious Freedom. J Relig Health (2022). https://doi.org/10.1007/s10943-022-01557-x. All in all, I find this revised article of importance to its field and I believe it fits well with the aims of IJERPH. For that reason  recommend its for publication. At the same time the papers requires some minor text editing.

Author Response

Dear Reviewer,

Thank you for your comments.

regarding the Abstract which contains some errors, i.e., line 63: the lowest/the highest: “the lowest acceptance rates were the highest in Oman (15.0%), Senegal (20.0%), Morocco (20.7%), and Jordan (8.5%)”. 

Done, it is now changed to:  In Facebook, the overall vaccine acceptance was 40.3%; the lowest acceptance rates were evident in Jordan (8.5%), Oman (15.0%), Senegal (20.0%) and Morocco (20.7%)

Also, line 68: “The differences in vaccine acceptance across countries and continents in Facebook and Twitter and were statistically significant”.

Done

Moreover, from reading abstract alone the readers can wonder what does ‘serious comments’ meant. Please rephrase. Done

Regarding the tone of the comments, in Facebook, countries that had the highest number of serious tone comments were Sweden (90.9%), United States (61.3%), and Thailand (58.8%). 

Finally, while referring to religious factors affecting VH the Authors might check recently published paper: Jones, D.G. Religious Concerns About COVID-19 Vaccines: From Abortion to Religious Freedom. J Relig Health (2022). https://doi.org/10.1007/s10943-022-01557-x.

Done added, thanks for the recommendation.